# Isavuconazole Therapy for Patients with Hematologic Diseases and Hematopoietic Cell Transplantation with and Without Breakthrough Invasive Fungal Infections

**DOI:** 10.3390/jof11090648

**Published:** 2025-09-01

**Authors:** Fabián Herrera, Diego Torres, Gustavo Mendez, Noelia Mañez, Rosana Jordán, Adriana Manzur, Myrna Cabral, Manuel Alderete, Natalia García Allende, José Benso, Claudia Salgueira, María Laura Pereyra, Hugo Peretti, Carla Niveyro, Maximiliano Castro, Federico Pollastrelli, Silvina García Rojas, Juan Dapás, Agustina Risso Patrón, Verónica Fernández, Rocío Gago, Javier Afeltra

**Affiliations:** 1Infectious Diseases Section, Internal Medicine Department, Centro de Educación Médica e Investigaciones Clínicas (CEMIC), Buenos Aires C1431, Argentina; diegots23@hotmail.com (D.T.); maxigabrielcastro@gmail.com (M.C.); 2Infectious Diseases Service, Hospital Dr. Ramón Madariaga, Misiones N3300, Argentina; mendez.doc@gmail.com (G.M.); caveyro@gmail.com (C.N.); 3Infectious Diseases Section, Internal Medicine Department, Hospital Italiano de Buenos Aires, Buenos Aires C1199ABB, Argentina; noelia.manez@hospitalitaliano.org.ar; 4Infectious Diseases Service, Hospital Británico de Buenos Aires, Buenos Aires C1280AEB, Argentina; rosanajordan61@gmail.com (R.J.); fedejuan88@gmail.com (F.P.); garcia@hbritanico.com.ar (S.G.R.); 5Infectious Diseases Service, Hospital Rawson, San Juan J5400, Argentina; manzuradriana@gmail.com; 6Infectious Diseases Service, Hospital Central, Mendoza M5502, Argentina; myrnacabral@hotmail.com (M.C.); juanidapas@hotmail.com (J.D.); 7Instituto Alexander Fleming, Buenos Aires C1426, Argentina; manuangel.alderete@gmail.com; 8Infectious Diseases Service, Hospital Alemán, Buenos Aires C1118AAT, Argentina; ngarciaallende@gmail.com (N.G.A.); agusrisso@hotmail.com (A.R.P.); 9Infectious Diseases Section, Internal Medicine Department, Hospital Italiano de San Justo, Buenos Aires B1754AZK, Argentina; jose.benso@hospitalitaliano.org.ar (J.B.); veronica.fernandez@hospitalitaliano.org.ar (V.F.); 10Infectious Diseases Service, Sanatorio Anchorena, Buenos Aires C1425, Argentina; clasalgueira@gmail.com; 11Infectious Diseases Service, Hospital Universitario Austral, Buenos Aires B1629, Argentina; laurapereyra@yahoo.com.ar (M.L.P.); rociogago1984@gmail.com (R.G.); 12Hematopoietic Stem Cell Transplant Unit, Sanatorio Británico, Rosario S2000, Argentina; perettihugoisidro@gmail.com; 13Microbiology Laboratory, Hospital Dr. José María Ramos Mejía, Buenos Aires C1221ADC, Argentina; javierafeltra@gmail.com

**Keywords:** isavuconazole therapy, hematologic diseases, hematopoietic cell transplantation, breakthrough invasive fungal infections

## Abstract

There are no data available on the effectiveness and safety of isavuconazole (ISA) for treating breakthrough invasive fungal infections (bIFIs). A retrospective and prospective cohort study was conducted between January 2020 and March 2025 in 13 centers in Argentina. Hematologic diseases (HD) and hematopoietic cell transplantation (HCT) patients who received ISA for IFI were included and followed for 12 weeks. Patients with proven and probable bIFIs and non-bIFIs were compared. One hundred and sixty-three patients were included. IFIs were classified as proven (13.5%), probable (26.9%) and possible (59.5%). Among 66 proven and probable IFIs, 53% were bIFIs, with aspergillosis and mucormycosis being the most common. Twenty-three (34.8%) patients had acute myelogenous leukemia, and 40.9% had received HCT. Forty-eight (72.7%) patients experienced neutropenia, with a median duration of 26 days (interquartile range [IQR] 16–44). Fluconazole and posaconazole were the most frequently received antifungal prophylaxis. ISA was prescribed as first-line therapy in 31 (46.9%) patients. The other 35 received ISA as a continuation therapy, mainly as a step-down therapy after liposomal amphotericin B. Four (6.1%) patients developed adverse effects, and one discontinued ISA. The 90-day overall clinical response between patients with bIFI vs. non-bIFI was 91.4% vs. 70.9% (*p* = 0.052). The 90-day overall and IFI-related mortality rates were, respectively, 11.4% vs. 32.3% (*p* = 0.068) and 5.7% vs. 9.7% (*p* = 0.659). The study data evidence ISA effectiveness and safety for the treatment of HD and HCT patients with and without bIFIs.

## 1. Introduction

Invasive fungal infections (IFIs) are a frequent complication in patients with hematologic diseases (HD) and hematopoietic cell transplantation (HCT), with significant morbidity and mortality rates, as well as high healthcare costs [1,2,3,4].

Patients with acute leukemia and prolonged neutropenia, as well as those with allogeneic HCT with high doses of corticosteroids, have an IFI incidence of 7–13.2% and 8.8–16%, respectively [1,2,3,5]. The epidemiology of IFIs has evolved worldwide, with a significant predominance of mold in recent decades, largely *Aspergillus* spp. *Mucorales* and *Fusarium* spp. [1,2]. Two multicenter studies have addressed this issue. Data from the TRANSNET surveillance study in the United States identified 983 IFIs among 875 HCT recipients, with invasive aspergillosis being the most common (43%) [6]. According to the Prospective Antifungal Therapy (PATH) Alliance registry, invasive aspergillosis was the most common IFI (59.2%) among 234 adult HCT patients [7]. More recently, a study carried out in Switzerland in 515 allogeneic HCT recipients showed that 48 (9.3%) patients developed 51 proven/probable IFI, with invasive aspergillosis (67%) and mucormycosis (18%) being the most frequent [8]. This is largely due to primary antifungal prophylaxis strategies, which are highly active against *Candida* spp. [9,10]. In this regard, all high-risk patients currently receive antifungal prophylaxis, even before this epidemiology change [11]. IDSA, ESCMID, ASTCT, ECIL, and AGIHO/DGHO guidelines recommend posaconazole (POSA) use as the first choice, followed by voriconazole (VORI). Echinocandins and fluconazole (FLUCO) are recommended as alternative drugs due to their narrow spectrum. Some guidelines consider isavuconazole (ISA) as an alternative antifungal prophylaxis in those cases where POSA and VORI are not appropriate (prolonged QTc, patients who receive QTc-prolonging medications, or drug–drug interaction) [12,13,14,15,16]. Therefore, most of the IFIs generally developed are breakthrough IFIs (bIFIs) [17]. In this sense, a systematic review and meta-analysis that identified 991 patients who received ISA prophylaxis found an incidence of bIFIs of 7% [18]. They pose a significant challenge for diagnosis; furthermore, no randomized studies have been conducted to determine the optimal treatment option [19,20].

In this complex scenario, several antifungal drugs have proved effective for treating IFIs. ISA was approved for the treatment of invasive aspergillosis and mucormycosis, based on the SECURE and VITAL trials [21,22]. After the implementation of these studies ISA was approved by key regulatory agencies, including the US Food and Drug Administration (FDA) and the European Medicines Agency (EMA) [23,24]. Given its efficacy and safety, several guidelines recommend ISA as a first-line treatment for invasive aspergillosis and as a step-down therapy or first-line therapy for mucormycosis [12,25]. In addition, many real-life studies are consistent with the data obtained in pivotal studies [26,27,28,29,30].

ISA has several advantages over VORI, including *Mucorales* coverage, stable drug levels with low intra- and interpatient variability, lower CYP3A4 inhibition, resulting in significantly fewer drug–drug interactions, a highly safe profile, and less drug discontinuation, among others [31,32,33,34,35,36]. In terms of adverse effects, clinical trial data show an overall tolerability profile comparable to voriconazole, with nausea, vomiting, and diarrhea being the most common events. However, ISA is associated with a lower incidence of hepatotoxicity and visual disturbances, with lack of QTc prolongation being a key differentiating safety feature. On the other hand, it causes mild concentration-dependent QTc interval shortening. It is also associated with fewer severe skin reactions compared to other azoles [21,22]. These characteristics are crucial in patients with HD and HCT, as they receive a large number of medications, especially immunosuppressants, which interact with potent CYP3A4 inhibitors, such as VORI. These inhibitors can also have a high rate of adverse effects and are therefore unsuitable for use in these patients.

To the best of our knowledge, no multicenter studies have been published despite all the above mentioned advantages of ISA therapy for bIFIs.

This study aimed to outline the use of ISA for the treatment of IFIs in patients with HD and HCT. We further aimed to describe and compare those who have proven and probable bIFI and non-bIFI in terms of ISA effectiveness and safety, and patients’ outcomes.

## 2. Materials and Methods

### 2.1. Setting, Patients and Study Design

A retrospective and prospective observational multicenter study was performed in 13 referral teaching centers (8 private and 5 public) specialized in the management of patients with HD and HCT in Argentina.

Adult patients (≥18 years of age) treated with ISA for IFIs and managed as inpatients or outpatients were included. The retrospective cohort comprised patients included from 1 January 2020 to 31 March 2024; however, all of them were treated and followed up by the Infectious Diseases physicians conducting the study. Patients from the prospective cohort were recruited from 1 April 2024 to 31 March 2025. For the total cohort, the following criteria were met: (a) patients presenting with HD or autologous and allogeneic HCT; (b) those treated with ISA for IFIs for at least 7 days; and (c) those followed until day 90 since the beginning of ISA or until the patient’s death, whichever occurred first.

Patients were excluded in case of missing data that precluded the assessment of baseline, clinical, microbiological, treatment characteristics, and outcomes.

Patients were identified through data files from the Infectious Diseases Services, which treat and follow up all patients with HD and HCT at each center. Data were obtained from direct patient care, medical records, and data from laboratory, microbiology, and pathology databases. Clinical, microbiological, diagnostic, treatment, and outcome variables from the total cohort were evaluated. In addition, these variables were compared between patients with proven and probable bIFIs and non-bIFIs.

Patient data were recorded with RedCap (Research Electronic Data Capture) software (RedCap version 13.7.19) and the server hosting was provided by the Argentine Society of Infectious Diseases.

The study was approved by the Ethics Committees from the different participating institutions, and patient informed consent was waived.

### 2.2. Definitions

Proven, probable, and possible IFIs were defined according to the revised and updated European Organization for Research and Treatment of Cancer and Mycoses Study Group EORTC/MSG criteria [37]. Proven IFI was defined as histopathologic, cytopathologic, direct microscopic examination, or culture of a biopsy or other specimen obtained by a sterile procedure from a normal sterile site. Probable mold infection was defined as that occurring in patients with (a) one host factor: recent history of neutropenia (<500 neutrophils/mm^3^) for >10 days, allogeneic HCT, prolonged use of corticosteroids, treatment with other T-cell or B-cell-immunosuppressants; (b) at least one clinical feature: pulmonary CT-scan showing nodules with or without a halo sign, air crescent sign, cavity, consolidation or a reverse halo sign; evidence of tracheobronchitis, sino-nasal or central nervous system infection; and (c) microbiological evidence: microscopic detection or culture of any mold from bronchoalveolar lavage (BAL) or sinus aspirates, galactomannan (GM) antigen test ≥ 1.0 from serum or BAL, or single serum or plasma ≥ 0.7 and BAL fluid ≥ 0.8 or *Histoplasma* urinary antigen. A possible mold infection was defined as that occurring in patients with one host factor and at least one pulmonary imaging on CT scan.

bIFIs were defined according to the Mycoses Study Group Education and Research Consortium (MSG-ERC) and the European Confederation of Medical Mycology (ECMM) [38]. bIFI was defined to occur during exposure to an antifungal drug, including fungi outside the spectrum of activity of an antifungal. bIFI time point is the first clinical sign or symptom, mycological finding, or radiological feature attributable to it. bIFI time point begins when each antifungal reaches plasma steady state and finishes during the last dose interval upon drug discontinuation.

Risk factors for IFIs were considered and defined as follows: (a) neutropenia < 500 neutrophils/mm^3^ for >10 days, and profound and prolonged neutropenia < 100 neutrophils/mm^3^ for >14 days prior to the diagnosis of IFI; (b) high doses of corticosteroids, such as prednisone (or equivalent) at doses ≥ 20 mg/day for a period ≥ 2 weeks prior to IFI, and the use of biological agents and/or anti-lymphocyte therapies within three months prior to IFI; (c) recent chemotherapy, such as the cycle of immunosuppressant drugs within one month prior to the diagnosis of IFI; (d) T-cell depletion, such as antithymocyte globulin or alemtuzumab for conditioning regimen of allogeneic HCT; (e) graft-versus-host disease (GvHD) and grading consistent with consensus guidelines [39]; (f) cytomegalovirus infection or disease occurring within 15 days prior to the diagnosis of IFI; (g) iron overload as ferritin serum level > 2000 ng/mL [40]; and (h) no HEPA filter system in the isolation room for induction chemotherapy in acute leukemia.

The first-line antifungal treatment was selected by the investigator based on the suspicion or diagnosis of IFI according to published guidelines [12,13,14,15,25]. Continuation treatment with ISA was considered either as a step-down therapy, or when the first-line drug could not be prescribed.

Preemptive therapy was given to patients whose diagnosis was based on positive GM and/or pulmonary CT-scan imaging, while targeted therapy implied that the diagnosis was made by microscopic detection or culture of any mold or yeast in clinical samples.

Favorable response to treatment was defined as absence of fever, improved signs and symptoms of the initial infectious source, decrease in GM index, and/or improved results in CT-scan imaging. Partial remission: improved signs and symptoms and imaging, though without resolution. Stable disease: improved signs and symptoms, though with no changes in imaging. Cure: clinical and imaging resolution.

Unfavorable outcome was defined as the patient’s death during the follow-up. Attributable mortality was considered to be the patient’s death with no response to treatment and documented clinical, radiological, microbiological, or histological findings suggestive of active IFI.

### 2.3. Statistical Analysis

The study population was characterized by descriptive statistics. For continuous variables, centrality (median) and dispersion (IQR) measures were used according to the distribution of variables. Categorical variables were analyzed using absolute frequency and percentage. Groups were compared using the U Mann–Whitney test for continuous variables and the Fisher exact test or the chi-square test for categorical variables. For all tests, a 95% level of statistical significance was used. Analyses were performed with the SPSS (Statistics for Windows, Version 22.0, Armonk, NY, USA) software packages.

## 3. Results

### 3.1. Characteristics and Outcomes of Patients’ Cohort

A total of 163 patients (126 retrospective and 37 prospective) diagnosed with IFI were included. Acute myelogenous leukemia (AML) and acute lymphoblastic leukemia (ALL) were the most frequent underlying diseases (96, 58.9%), and were active in 125 (76.6%). Forty-nine (30.1%) had undergone HCT, with allogeneic being the most common type. Thirteen (43.3%) and 7 (23.3%) of allogeneic HCT developed acute and chronic GvHD, respectively. From the total cohort, 63 (38.6%) patients presented with oral mucositis, and 10 (6.1%) HCT patients developed acute or chronic GvHD affecting the gastrointestinal tract. One hundred and thirty-two (80.9%) patients were neutropenic at the onset of IFI. Of them, 115 (87.1%) presented profound and prolonged neutropenia, with a median duration of 27 days (IQR: 16–47). The cohort comprised patients with increased risk factors for IFIs, with a median of 2 (IQR: 2–3). Patients’ baseline characteristics and risk factors are outlined in Table 1 and Figure 1.

Proven and probable IFIs were diagnosed in 66 (40.5%) patients and possible IFIs in 97 (59.5%). Ninety-two (56.4%) were bIFIs. The most common locations were lungs (147, 90.2%) and paranasal sinuses (23, 14.1%). Lung CT scan showed nodules in 89 (54.6%) and halo sign in 37 (22.7%). IFIs characteristics are depicted in Table 2.

ISA treatment was prescribed as preemptive therapy in 136 (83.4%) and as targeted therapy in 29 (17.8%) patients. It was used as first-line therapy in 73 (44.8%) of all the patients, and 153 (93.8%) received it as monotherapy. Ten (6.1%) patients received ISA in combination with L-AmB. The other antifungal drugs used as first-line therapy (90 patients) were L-AmB (65, 72.2%), VORI (13, 14.4%), POSA (6, 6.7%), LC-AmB (4, 4.4%), caspofungin (1, 1.1%), and FLUCO (1, 1.1%). ISA administration route was oral (77, 47.2%), intravenous (IV) followed by oral (57, 34.9%), IV (27, 16.6%), and oral followed by IV (2, 1.2%), with a median duration of 90 days (IQR: 59–113). Five patients undergoing treatment with medication that interacts with POSA and VORI (venetoclax 2, sirolimus 2 and ponatinib 1) were therefore administered ISA.

Seven (4.3%) patients developed related adverse effects (nausea 1, hepatobiliary abnormalities 6, rash 1, and shortened QTc 1), and only 1 (0.6%) had to discontinue ISA. One hundred and thirty (79.7%) patients achieved a favorable response at week 12 (cure 54.6%, partial remission 19.6%, and stable disease 5.5%). Overall mortality was 22.7%, and IFI-related mortality was 4.9%. Ninety-day overall mortality in patients with pulmonary vs. non-pulmonary locations was 23.2% vs. 18.4% (*p* = 0.53).

### 3.2. Characteristics and Outcomes of Patients with Probable and Proven IFIs

A total of 66 patients were diagnosed with proven and probable IFIs, and 35 (53%) of them were bIFIs. Both bIFIs and non-bIFIs patients had similar baseline characteristics regarding sex, Charlson score comorbidity index, underlying diseases, and disease status. Forty-eight (72.7%) patients had neutropenia, mostly profound and prolonged, with a median duration of 26 days (IQR: 16–44).

The most common primary antifungal prophylaxis administered to bIFIs patients was FLUCO (13, 37.1%), followed by POSA (10, 28.6%). Baseline characteristics and the antifungal prophylaxis prescribed are outlined in Table 3.

Invasive aspergillosis and mucormycosis were the most frequent IFIs in both groups. We compared and contrasted IFI locations in non-bIFI and bIFI patients and found lungs in 26 (83.9%) vs. 27 (77.1%), *p* = 0.492, and paranasal sinuses in 7 (22.6%) vs. 8 (22.9%), *p* = 0.97, respectively. A few patients had other locations. The most common CT-scan findings were nodules in 32 (48.5%) patients, halo sign in 17 (25.8%), and a ground glass appearance in 20 (30.3%), with no differences between groups. IFIs were diagnosed by microscopic detection in 16 (24.2%) patients, culture in 29 (43.9%), GM test in 35 (53%), and histopathology in 17 (25.8%).

The etiology and methodology used for the diagnosis of proven and probable IFIs are described in Table 4.

ISA was prescribed as monotherapy in 62 (93.9%) patients and as first-line treatment in 31 (46.9%). In the bIFI group, 15 patients (42.9%) received ISA as first-line treatment. Ten (66.7%) were under prophylaxis with FLUCO, 2 (13.3%) with VORI, 2 (13.3%) with L-AmB, and 1 (6.7%) with LC-AmB. As a continuation treatment, it was used in 20 patients (57.1%). The reasons for prescribing ISA were step-down therapy after L-AmB in 11 (55%), (10 had undergone POSA prophylaxis), and VORI in 1 (5%); related-adverse effects with L-AmB in 2 (10%) or VORI in 2 (10%); and combination treatment with L-AmB in 3 (15%), or caspofungin in 1 (5%). The median duration of treatment in bIFI vs. non-bIFI patients was 90 days (IQR: 62.5–119.5) vs. 85 days (IQR: 61.5–149), *p* = 0.908. Source control was performed in 18 (27.3%) patients, and mostly consisted of paranasal sinus endoscopic surgery (7 patients in each group). Treatment of proven and probable characteristics is described in Table 5.

Regarding outcomes, a large number of patients achieved a favorable response, being higher in those with bIFI. Overall and IFI-related mortality in this group were 11.4% and 5.7%, respectively. Outcome variables are shown in Figure 2.

## 4. Discussion

The study assessed the effectiveness and safety of ISA for treating HD and HCT patients with IFIs. Patients with several risk factors for IFIs, most of them neutropenic, were included. Patients with proven and probable IFIs, either non-bIFIs or bIFIs, were analyzed separately. Aspergillosis and mucormycoses were the IFIs most commonly diagnosed, largely located in the lungs and paranasal sinuses. Almost half of the cases received ISA as a first-line treatment, mainly as monotherapy. Among bIFI patients, those who received ISA as a first-line therapy were mainly undergoing FLUCO or VORI prophylaxis. Half of the patients treated with ISA as a continuation therapy after L-AmB were receiving POSA prophylaxis. In a large proportion of patients with bIFIs, a clinical response was observed, with low overall and IFI-related 90-day mortality rates.

These are the major findings of the present study: first, in terms of IFI epidemiology, our cohort had similarities and differences with other real-life studies. In a multicenter study including centers from the USA, Europe, and Brazil, aspergillosis was the leading cause of IFI (79%), followed by fusariosis (8%) [41]. In contrast, in our cohort fusariosis only represents 1.5%. Second, approximately 50% of the patients had bIFI. Aspergillosis and mucormycoses were the most frequent molds in those with proven and probable IFI. In this respect, a multicenter study from Spain reported bIFI in 121 cases, with aspergillosis in 59% and mucormycosis in 7% being the most common among 94 patients with proven or probable bIFI [42]. Third, as reported in the SECURE trial and many real-life studies, most IFI locations were the lungs [21,26,27,29].

Fourth, bIFIs are currently and frequently a major concern in clinical practice among HD and HCT patients, as it is difficult to identify the type of fungus involved. The sensitivity of GM in BAL proved to be lower in patients that receive antifungal prophylaxis compared to those that do not: 52% vs. 81%. Likewise, the sensitivity of serum GM and culture in BAL is 31.3% and 18.8%. A combination of these methods can increase diagnostic efficacy [43,44]. Based on these findings, in our cohort bIFI could be diagnosed using several diagnostic methods. In this regard, the ECMM consensus status recommends the use of all available methods to diagnose bIFI [45]. Fifth, almost half of the patients received ISA as a first-line treatment, mostly as monotherapy. This differs from the literature, which recommends using L-AmB as a first-line therapy [19,20,46].

Sixth, 10 patients under continuation ISA therapy after L-AmB were receiving primary prophylaxis with POSA. Interestingly, ISA and POSA have the same antifungal spectrum, and both proved effective for aspergillosis and mucormycosis [21,22,47]. Antifungal cross-resistance between these two azoles does not necessarily occur. Cross-resistance between POSA (used for prophylaxis) and ISA (used for treatment) is complex and not fully deterministic. While a theoretical risk exists due to their shared drug class, several factors mitigate absolute cross-resistance. Thus, ISA could be considered a viable therapeutic option even after POSA prophylaxis failure. In this sense, the binding affinity of ISA to the fungal Cyp51A enzyme target is different from that of POSA. The environmentally driven tandem repeats (TR34/L98H and TR46/Y121F/T289A) are the most relevant resistance mechanisms [48]. These mutations often confer panazole resistance to both POSA and ISA. However, their prevalence is not yet universal, and breakthrough infections in patients on prophylaxis can still be caused by wild-type or susceptible isolates with other resistance mechanisms. Key studies support this lack of absolute cross-resistance. The phase 3 SECURE trial and subsequent analyses have documented successful outcomes with ISA in patients who had received prior azole prophylaxis, including POSA [21].

Moreover, one of the several reasons for developing bIFI is that the antifungal agent does not achieve enough serum levels. This is frequently observed in VORI, but has also been described with POSA, even in tablet formulations. A study in patients with acute leukemia and HCT receiving POSA prophylaxis found that 18% of them had subtherapeutic serum levels (<700 ng/mL). Factors such as having diarrhea, receiving proton pump inhibitors, and weighing more than 90 kg were associated with subtherapeutic serum levels, with the first two being common in HD and HCT patients [49]. Nonetheless, as POSA serum levels were not available, we cannot state that this could occur in our patients. On the other hand, ISA levels are adequate, even in patients with mucositis and gastrointestinal GvHD, as could often be the case with many of our patients [50,51]. Seventh, the cohort had a low rate of related adverse effects and drug discontinuation, which is consistent with the literature [29,30]. Given its safety, ISA is suitable for treating IFIs in severely ill patients.

Eighth, our patients with bIFI had higher clinical response and lower mortality rate than those in other real-life studies, which report a mortality rate of 35% [17]. Apart from the effectiveness of ISA, 62.9% of our patients received preemptive therapy, which means that they were treated early. Several studies have demonstrated that this strategy is consistent with higher survival rates [52,53,54]. We also consider that other factors could have contributed to the high clinical response and low mortality rate, particularly in patients with bIFI. They were younger than non-bIFI patients and had a higher rate of underlying disease in complete or partial remission. Moreover, since all the cohorts were diagnosed with IFI and were followed up by the ID physicians participating in the study, we assume that the diagnostic and therapeutic approach has been timely and appropriate.

Our study has some limitations that should be considered. First, data on the azoles serum levels or susceptibility testing were not available. Patients’ good outcomes could only be partly explained by these factors. However, there are no clinical susceptibility cut-off values for molds, except for *Aspergillus fumigatus* complex. In addition, the association between exposure to and efficacy of ISA treatment has not been proven [55]. Second, the sample size of patients with bIFI was small. Therefore, a larger population is required to evaluate the outcome variables. Third, several patients were retrospectively included, which may have led to some biased results. Notwithstanding that, all those patients were treated and prospectively followed by the Infectious Diseases physician conducting the study and missing data were not allowed.

The strengths of our study rely on its multicenter design. It was carried out in healthcare facilities specialized in the treatment of patients with HD and HCT. In addition, all of them were evaluated, treated, and followed up by the investigators. Therefore, our results accurately reveal IFIs complex scenario. Moreover, our study comprised the largest cohort from Latin America.

In conclusion, the study data evidenced ISA effectiveness and safety for the treatment of HD and HCT patients with IFI. It further showed a suitable option for treating patients with bIFIs. However, larger studies should be conducted to confirm this finding.

## Figures and Tables

**Figure 1 jof-11-00648-f001:**
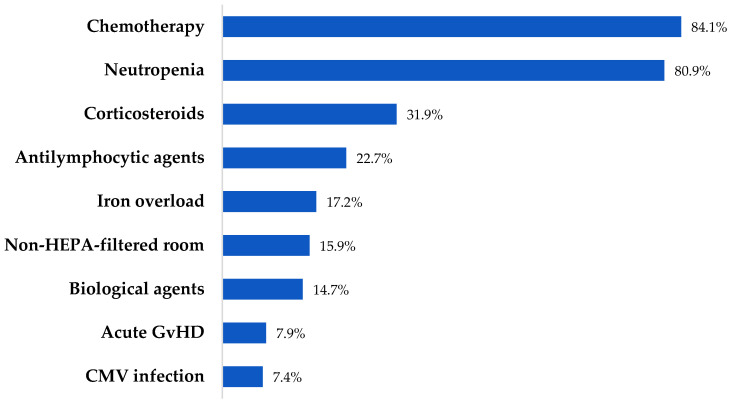
Risk factors for IFIs.

**Figure 2 jof-11-00648-f002:**
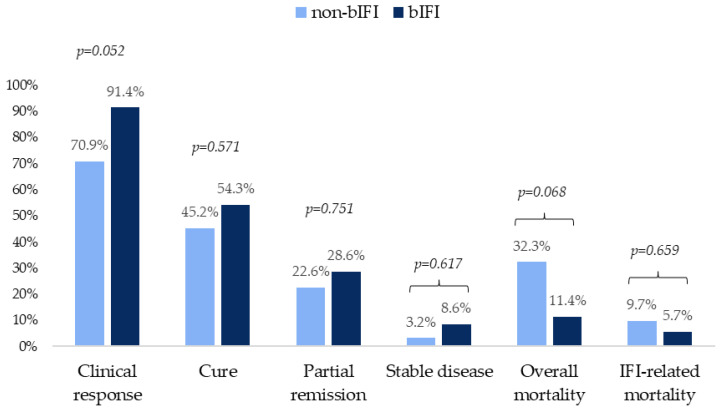
Clinical outcomes at 12 weeks in patients with probable and proven non-bIFIs and bIFIs.

**Table 1 jof-11-00648-t001:** Baseline characteristics of patients diagnosed with IFIs.

Variables	Patients (*n* = 163) N (%)
Age (years) median (IQR)	50 (38–63)
Sex, male	95 (58.3)
Charlson comorbidity index score ≥ 3	73 (44.8)
Underlying disease	
Acute myelogenous leukemia	76 (46.6)
Acute lymphoblastic leukemia	20 (12.3)
Non-Hodgkin lymphoma	16 (9.8)
Myelodysplastic syndrome	8 (4.9)
Hodgkin lymphoma	11 (6.7)
Aplastic anemia	9 (5.5)
Chronic myelogenous leukemia	6 (3.7)
Chronic lymphoblastic leukemia	4 (2.4)
Multiple myeloma	7 (4.3)
Others	6 (3.7)
Disease status	
Complete remission	38 (23.3)
Partial remission	8 (4.9)
Relapsed	33 (20.2)
Refractory	18 (11)
Recently diagnosed	66 (40.5)
HCT	49 (30.1)
Allogeneic	30 (18.4)
HLA matching and donor type	
Haploidentical	15 (9.2)
Matched unrelated donor	5 (3.1)
Matched related donor	10 (6.1)
T-cell depletion	9 (5.5)
Acute GvHD	13 (7.9)
Grade I	2 (1.2)
Grade II	4 (2.4)
Grade III	7 (4.3)
Grade IV	0 (0)
Chronic GvHD	7 (4.3)

Abbreviations: HCT = Hematopoietic Cell Transplantation; GvHD = Graft-versus-Host Disease. Other underlying diseases include chronic myeloid leukemia, myelofibrosis, hairy cell leukemia, Richter’s syndrome, and Sézary syndrome.

**Table 2 jof-11-00648-t002:** IFIs classification, location and radiological characteristics.

Variables	Patients (*n* = 163) N (%)
IFI classification	
Proven	22 (13.5)
Probable	44 (26.9)
Possible	97 (59.5)
IFI location	
Lungs	147 (90.2)
Paranasal sinuses	23 (14.1)
Liver	5 (3.1)
Skin and soft tissue	5 (3.1)
Central nervous system	3 (1.8)
Disseminated	3 (1.8)
Lung CT scan	
Nodules	89 (54.6)
Ground glass appearance	59 (36.2)
Halo sign	37 (22.7)
Tree in bud	22 (13.5)
Alveolar infiltrate	14 (8.6)
Cavity	3 (1.8)
Air crescent sign	1 (0.6)
Reverse halo sign	1 (0.6)

**Table 3 jof-11-00648-t003:** Baseline characteristics and primary antifungal prophylaxis of patients with proven and probable non-bIFIs and bIFIs.

Variable	Total (*n* = 66)	Non-bIFI (*n* = 31)	bIFI (*n* = 35)	*p*-Value
Age, median (IQR)	47 (39–61)	54 (43–67)	44 (36–52)	0.014
Male sex–n (%)	40 (60.6)	16 (51.6)	24 (68.6)	0.159
Charlson comorbidity index ≥ 3–n (%)	29 (43.9)	17 (54.8)	12 (34.3)	0.093
Underlying disease–n (%)				
Acute myelogenous leukemia	23 (34.8)	11 (35.5)	12 (34.3)	0.918
Acute lymphoblastic leukemia	7 (10.6)	1 (3.2)	6 (17.1)	0.41
Myelodysplastic syndrome	7 (10.6)	2 (6.4)	5 (14.3)	0.433
Non-Hodgkin lymphoma	4 (6.1)	4 (12.9)	0 (0)	0.043
Hodgkin lymphoma	10 (15.1)	4 (12.9)	6 (17.1)	0.738
Multiple myeloma	7 (10.6)	5 (16.1)	2 (5.7)	0.239
Disease status–n (%)				
Complete remission	16 (24.2)	5 (16.1)	11 (31.4)	0.165
Partial remission	4 (6.1)	1 (3.2)	3 (8.6)	0.616
Relapsed	17 (25.8)	9 (29)	8 (22.9)	0.567
Refractory	9 (13.6)	7 (22.6)	2 (5.7)	0.071
Recently diagnosed	20 (30.3)	9 (29)	11 (31.4)	0.832
HCT–n (%)	27 (40.9)	9 (29)	18 (51.4)	0.064
Allogeneic	13 (19.7)	4 (12.9)	9 (25.7)	0.228
Corticosteroid use–n (%)	24 (364)	11 (35.5)	13 (37.1)	0.888
Biological agents–n (%)	15 (22.7)	7 (22.6)	8 (22.9)	0.978
Antilymphocyte drugs–n (%)	12 (18.2)	3 (9.7)	9 (25.7)	0.117
Neutropenia–n (%)	48 (72.7)	21 (67.7)	27 (77.1)	0.392
Antifungal prophylaxis–n (%)	35 (53)	0 (0)	35 (100)	–
Fluconazole	13 (19.7)	0 (0)	13 (37.1)	–
Posaconazole	10 (15.1)	0 (0)	10 (28.6)	–
Voriconazole	2 (3)	0 (0)	2 (5.7)	–
L-AmB	5 (7.6)	0 (0)	5 (14.3)	–
LC-AmB	2 (3)	0 (0)	2 (5.7)	–
Caspofungin	3 (4.5)	0 (0)	3 (8.6)	–

Abbreviations: IFI, Invasive Fungal Infection; non-bIFI, non-breakthrough IFI; bIFI, breakthrough IFI; IQR, interquartile range; HCT, hematopoietic cell transplantation; L-AmB, Liposomal Amphotericin B; LC-AmB, Amphotericin B lipid complex. *p*-values were obtained using the chi-square or Fisher’s exact test for categorical variables and the Mann–Whitney U test for continuous variables.

**Table 4 jof-11-00648-t004:** Type of mycoses, etiology, diagnostic methods, and treatment of proven and probable non-bIFIs and bIFIs.

Variable	Total (*n* = 66)	Non-bIFI (*n* = 31)	bIFI (*n* = 35)	*p*-Value
Type of IFI–n (%)				
Aspergillosis	43 (65.1)	22 (70.9)	21 (60)	0.351
Mucormycosis	8 (12.1)	2 (6.4)	6 (17.1)	0.265
Fusariosis	1 (1.5)	0 (0)	1 (2.9)	1
Other hyalo or phaeohyphomycosis	5 (7.6)	2 (6.4)	3 (8.6)	1
Histoplasmosis	1 (1.5)	1 (2.9)	0 (0)	0.470
Cryptococcosis	1 (1.5)	1 (2.9)	0 (0)	0.470
Unidentified hyphae	8 (12.1)	3 (9.7)	5 (14.3)	0.713
Microscopic detection–n (%)	16 (24.2)	9 (29)	7 (20)	0.245
Septate branched hyphae	10 (15.1)	7 (22.6)	3 (8.6)	0.170
Coenocytic hyphae	5 (7.6)	1 (3.2)	4 (11.4)	0.360
*Cryptococcus* yeast	1 (1.5)	1 (3.2)	0 (0)	0.470
Culture isolates–n (%)				
*Alternaria* sp.	1 (1.5)	0 (0)	1 (2.9)	1
*Aspergillus* sp.	3 (4.5)	1 (3.2)	2 (5.7)	1
*Aspergillus flavus* complex	5 (7.6)	2 (6.4)	3 (8.6)	1
*Aspergillus fumigatus* complex	5 (7.6)	4 (12.9)	1 (2.9)	0.178
*Aspergillus niger*	3 (4.5)	2 (6.4)	1 (2.9)	0.596
*Cryptococcus neoformans* var. *neoformans*	1 (1.5)	1 (2.9)	0 (0)	0.470
*Cunninghamella* sp.	1 (1.5)	0 (0)	1 (2.9)	1
*Curvularia* sp.	2 (3)	2 (6.4)	0 (0)	0.216
*Fusarium* sp.	1 (1.5)	0 (0)	1 (2.9)	1
*Penicillium* sp.	2 (3)	0 (0)	2 (5.7)	1
*Rhizopus* sp.	2 (3)	0 (0)	2 (5.7)	1
*Rhizopus microsporum*	1 (1.5)	0 (0)	1 (2.9)	1
*Rhizopus oryzae*	1 (1.5)	0 (0)	1 (2.9)	1
*Rhizopus arrhizus*	1 (1.5)	0 (0)	1 (2.9)	1
*Aspergillus* GM–n (%)				
Positive in serum	12 (18.2)	5 (16.1)	7 (20)	0.684
Positive GM in BAL	17 (25.8)	9 (29)	8 (22.9)	0.567
Positive serum + BAL GM	6 (9.1)	3 (9.7)	3 (8.6)	1
*Histoplasma* urinary antigen–n (%)	1 (1.5)	1 (2.9)	0 (0)	0.470
Histopathology–n (%)				
Hyphal invasion of blood vessels	7 (10.6)	3 (9.7)	3 (8.6)	1
Septate branched hyphae	5 (7.6)	3 (9.7)	2 (5.7)	0.659
Coenocytic hyphae	6 (9.1)	2 (6.4)	4 (11.4)	0.676
Yeast	1 (1.5)	1 (3.2)	0 (0)	0.470

Abbreviations: IFI, Invasive Fungal Infection; non-bIFI, non-breakthrough IFI; bIFI, breakthrough IFI; GM, galactomannan; BAL, bronchoalveolar lavage. *p*-values were obtained using the chi-square or Fisher’s exact test for categorical variables.

**Table 5 jof-11-00648-t005:** Treatment of proven and probable non-bIFIs and bIFIs.

Variable	Total (*n* = 66)	Non-bIFI (*n* = 31)	bIFI (*n* = 35)	*p*-Value
ISA treatment modality-n (%)				
Preemptive therapy	39 (59.1)	17 (54.8)	22 (62.9)	0.508
Targeted therapy	29 (43.9)	14 (45.2)	15 (42.8)	0.851
Treatment with ISA–n (%)				
As first-line therapy	31 (46.9)	16 (51.6)	15 (42.9)	0.477
As continuation therapy	35 (53%)	15 (48.4)	20 (57.1)	0.477
Other first-line therapy–n (%)				
L-AmB	23 (34.8)	8 (25.8)	15 (42.9)	0.147
LC-AmB	1 (1.5)	1 (3.2)	0 (0)	0.470
Voriconazole	11 (16.7)	8 (25.8)	3 (8.6)	0.097
Posaconazole	1 (1.5)	1 (3.2)	0 (0)	0.470
Fluconazole	1 (1.5)	1 (3.2)	0 (0)	0.470
Caspofungin	1 (1.5)	0 (0)	1 (2.9)	1

Abbreviations: IFI, Invasive Fungal Infection; non-bIFI, non-breakthrough IFI; bIFI, breakthrough IFI; ISA, isavuconazole; L-AmB, Liposomal Amphotericin B; LC-AmB, Amphotericin B lipid complex. *p*-values were obtained using the chi-square or Fisher’s exact test for categorical variables.

## Data Availability

Data are available upon request. Contact the corresponding author.

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
