# Peer review of "Isavuconazole Therapy for Patients with Hematologic Diseases and Hematopoietic Cell Transplantation with and Without Breakthrough Invasive Fungal Infections"

_jof, 2025, doi:10.3390/jof11090648_

Round 1

Reviewer 1 Report

Herrera and colleagues submit a lookback study for HCT and heme malignancy patients from 13 Argentinian medical centers undergoing isavuconazole therapy who may or may not develop breakthrough IFI infections.

Comments:

  • To this reviewer, the biggest problem with the paper is the addition of possible infections. It’s just so unclear if they are truly fungal infections. It’s almost as if they should be examined completely separately. However, I understand that many papers are published with possible IFI patients.
  • The usual abbreviation for hematopoietic stem cell transplantation (HSCT) has evolved over the years, and it is currently hematopoietic cell transplantation (HCT).
  • The sentence: “This phenomenon is mainly due to the use of primary antifungal prophylaxis strategies, which are highly active against Candida spp.”, needs a reference. You have many choices. 
  • The sentence: “In this regard, all high-risk patients currently receive antifungal prophylaxis.”, needs a reference. You have many choices, with examples including PMID 38481428, PMID 31688884, PMID 31507152, PMID 29860307, etc.
  • The phrase “as they receive a large number of drugs” would be more appropriate for a manuscript if replaced with “as they receive a large number of medications”.
  • This sentence can be improved, from “Preemptive therapy consisted of positive GM and/or pulmonary CT-scan imaging, while targeted therapy implied microscopic detection or culture of any mold or yeast in clinical samples” to “Preemptive therapy was given to patients whose diagnosis was based on positive GM and/or pulmonary CT-scan imaging, while targeted therapy implied that the diagnosis was made by microscopic detection or culture of any mold or yeast in clinical samples.”
  • It is ok to stop percentages to zero or one decimal point (change 87.12% to 87% or 87.1%).

Author Response

Thank you for all your comments. I´ve attached the responses and submitted a new version of the manuscript. 

Reviewer 2 Report

The paper is a multicenter study describing their experience about the effectiveness of isavuconazole in patients suffering from Invasive Fungal Infections either breakthrough or not. In general it is an interesting and well presented paper. Some corrections in the beginning are necessary which I am presenting downwards. 

Some comments are included in the file , but they are few.

Author Response

(The authors gave the same response as above.)

Reviewer 3 Report

Dear Authors,

I read your manuscript addresses an important and timely topic: the effectiveness and safety of isavuconazole in patients with hematologic diseases (HD) and hematopoietic stem cell transplantation (HSCT), with or without breakthrough invasive fungal infections (bIFIs). The multicentre design and relatively large cohort strengthen the study’s relevance as well as real-life data on ISA. However, the manuscript requires substantial revisions to improve clarity, structure, and completeness of the analyses and to clarify some points.

Introduction

  • The introduction is not fluent and lacks sufficient references and precise information. It should be revised to:
  • Provide key regulatory information (e.g., FDA/EMA approval date of isavuconazole).
  • Include updated guideline recommendations (ECIL, ESCMID, IDSA).
  • Expand epidemiological data on IFIs in HD and HSCT patients.
  • Summarise the spectrum and adverse events of isavuconazole compared to other antifungals.
  • The discussion should later be structured in parallel with this improved introduction for coherence.
  • I suggest reading and citing:

Khatri, A. M., Natori, Y., Anderson, A., Jabr, R., Shah, S. A., Natori, A., Chandhok, N. S., Komanduri, K., Morris, M. I., Camargo, J. F., & Raja, M. (2023). Breakthrough invasive fungal infections on isavuconazole prophylaxis in hematologic malignancy & hematopoietic stem cell transplant patients. Transplant infectious disease : an official journal of the Transplantation Society25 Suppl 1, e14162. https://doi.org/10.1111/tid.14162

Keiko Ishida, Mizuki Haraguchi, Muneyoshi Kimura, Hideki Araoka, Akina Natori, John M Reynolds, Mohammed Raja, Yoichiro Natori, Incidence of Breakthrough Fungal Infections in Patients With Isavuconazole Prophylaxis: A Systematic Review and Meta-analysis, Open Forum Infectious Diseases, Volume 12, Issue 4, April 2025, ofaf163, https://doi.org/10.1093/ofid/ofaf163

Caitlin R Rausch, Adam J Dipippo, Dimitrios P Kontoyiannis, Breakthrough Invasive Fungal Infections (bIFI) in Adult Patients with Leukemia Receiving Isavuconazole (ISA), Open Forum Infectious Diseases, Volume 4, Issue suppl_1, Fall 2017, Page S718, https://doi.org/10.1093/ofid/ofx163.1931

Methods

  • Lines 107–109 (lack of antifungal serum levels and susceptibility testing) should be moved to the Limitations section.
  • Please specify the software version of RedCap used.
  • In Table 1, clarify what is included in the “Others” category.
  • Patients should be stratified not only by underlying hematologic disease but also by treatment regimens (chemotherapy based on?). This analysis is critical to evaluate drug safety and possible interactions.
  • Figure 1 should report specific chemotherapy regimens, not just “chemotherapy” as a general risk factor.

If patient heterogeneity required non-parametric tests, please provide these analyses, possibly as a supplementary file.

Results

  • Provide explicit outcome data in lines 260–261 instead of a narrative statement.
  • Report detailed data on skin and soft tissue infections and include clinical outcomes stratified by site of infection (lung, sinus, soft tissue, CNS, etc.).
  • Include an additional analysis stratifying patients by hematologic treatment regimen, as suggested above.

Discussion

The discussion, similar to the introduction, lacks fluency and structured argumentation. It should be revised to:

  • Confute and contextualise the results with respect to pivotal trials (SECURE, VITAL) and real-world studies (US, Europe, Latin America).
  • Expand on the issue of cross-resistance between posaconazole and isavuconazole, particularly in patients receiving posaconazole prophylaxis.
  • Better interpret clinically relevant but non-significant findings (e.g., trends in mortality reduction).

I also suggest highlighting the originality of the study (e.g., one of the first multicenter in Sud America)

Author Response

(The authors gave the same response as above.)

Round 2

Reviewer 1 Report

Herrera and colleagues submit a revised manuscript to address HCT and heme malignancy patients from 13 Argentinian medical centers undergoing isavuconazole therapy who may or may not develop breakthrough IFI infections. They have done a fine job in responding to the reviewers’ points.

  • In the author listing, why is “Multicenter study on” underlined?
  • In the author listing, “Isavuconazol” is missing an “e” at the end of the word.
  • In the first paragraph of the introduction, “Equinocandins” is spelled wrong. It should be “Echinocandins”.
  • In the first paragraph of the introduction, “QTc-prolong medications” is not exactly how an English-speaking person would write it. It could be “QTc-prolonging medications”.
  • The new sentence, “Patients with acute leukemia and prolonged neutropenia, as well as those with allogeneic HCT with high doses of corticosteroids, have an IFI incidence of 7-13.2% and 8.8-16%, respectively (1,2,3,5)”, has the references in parentheses rather than brackets. Have these actually been picked up by the reference manager (I suspect not), and as such are they the references that the authors intends?
  • The second paragraph of the discussion is quite lengthy. I realize that the authors are using this paragraph to present their major findings, but they could break it up into several paragraphs.

Author Response

We thank you for all your comments. We corrected all mistakes.

Reviewer 3 Report

Dear Authors,

All the corrections have been made, and the manuscript has been improved.

Dear Authors,

All the corrections have been made, and the manuscript has been improved.

Author Response

Dear reviewer,

We thank you for your help in improving our manuscript.